# Highly Efficient Adsorption of Sr^2+^ and Co^2+^ Ions by Ambient Prepared Alkali Activated Metakaolin

**DOI:** 10.3390/polym14050992

**Published:** 2022-02-28

**Authors:** Yi-Hsuan Huang, Yu-Chun Wu

**Affiliations:** Department of Resources Engineering, National Cheng Kung University, Tainan 701, Taiwan; k0988861297@gmail.com

**Keywords:** metakaolin, strontium, cobalt, alkali activation, adsorption, competitive adsorption, mineralization

## Abstract

This study aimed to explore a low cost and sustainable adsorbent to remove Sr^2+^ and Co^2+^ ions, which are major radioactive ions in nuclear wastewater. The material properties of the alkali-activated metakaoline as a function of soaking time at ambient temperature from 1 day to 7 days were examined by XRD, XRF, SEM, and solid-state NMR. Adsorption isotherms were used to evaluate the appropriate soaking time for the optimal sorption performance for both Sr^2+^ and Co^2+^ ions. The alkali-activated metakaolin soaked for 3 days (BK3) presented the maximum adsorption capacities of 3.81 meq/g (167.5 mg/g) and 4.02 meq/g (118.5 mg/g) for Sr^2+^ and Co^2+^, respectively. The sorption mechanisms for Sr^2+^ and Co^2+^ in the BK3 sample were investigated, and the experimental results indicated that adsorption for Sr^2+^ was achieved via ion exchange. By contrast, surface complexation in combination with ion exchange contributed to the sorption mechanisms for the removal of Co^2+^. Competitive adsorption experiments revealed that the alkali-activated metakaolin favored the adsorption for divalent ions (i.e., Sr^2+^ and Co^2+^), and it was less effective for Cs^+^. Finally, the used adsorbent could be directly mineralized and vitrified by heat treatment to immobilize the Sr^2+^ and Co^2+^ ions.

## 1. Introduction

Nuclear power is considered an efficient method of energy production, but a sustainable solution for treating the radioactive waste is still an unsolved issue. ^137^Cs^+^ and ^90^Sr^2+^ are the most abundant nuclear fission products, and ^60^Co^2+^ is the product from neutron activation in a nuclear reactor and must be removed from wastewater before being released into seawater [1]. Ion exchange resins are one of the most used adsorbents for the decontamination of industrial wastewater; however, their relatively low chemical and thermal stabilities may result in secondary problems when used as absorbents for removing radioactive ions. Natural zeolites, such as mordenite and clinoptilolite, are recognized as good adsorbents for the removal of Cs^+^ from radioactive wastewater; however, their sorption capacities to Sr^2+^ are relatively low [2,3]. In comparison with natural zeolites, synthetic zeolites prepared via the hydrothermal reaction in the NaOH solution present a better absorption capacity due to the effective ion exchange between Na^+^ and target ions [4,5,6,7]. Chen et al. synthesized Faujasite-type zeolite (FAU) via the hydrothermal reaction using metakaolin and fly ash as the starting materials mixed with NaOH solution, and found that FAU exhibits excellent adsorption capacities for Cs^+^, Sr^2+^, and Co^2+^ ions [8]. The hydrothermal reaction is limited to small batch production, and a suitable method that allows for mass production is essential for practical application in treating voluminous wastewater. Alkali activation is known as an approach to obtain zeolites under ambient pressure, and it is adaptable for mass production [9,10]. Alkali activation is used for preparing geopolymers using aluminosilicate-based precursor materials, such as metakaolin, fly ashes, and slag, and it can be use to remove heavy metal ions [10,11,12]. 

The present work aimed to explore a sustainable adsorbent for removing Sr^2+^ and Co^2+^ from nuclear wastewater. Metakaolin was selected as a precursor due to its abundance in the Earth. Instead of the hydrothermal reaction, alkali activation treatment by NaOH solution was carried out at ambient temperature in this work, indicating its potential for low-cost mass production without the need for a heating facility. The structural and compositional variations during the alkali activation process were characterized. The isotherm and kinetics sorption behaviors in correlation with the polymerization degree were investigated to understand the sorption mechanisms of Sr^2+^ and Co^2+^ ions. The selective sorption among Cs^+^, Sr^2+^, and Co^2+^ was also discussed. Finally, mineralization to the wasted adsorbents was attempted, and a preliminary leaching test was performed. 

## 2. Materials and Methods

### 2.1. Material Preparation

Metakaolin was used as precursor by heating natural kaolinite (purchased from the American Clay Minerals Society, Warren County, GA, USA) at 800 °C for 2 h. The as-obtained metakaolin was placed in a 6 M NaOH aqueous solution with a fixed solid/liquid ratio of 1/25 g/mL. Polymerization was conducted under agitation at ambient temperature for 1, 3, and 7 days. The alkali-activated powders were washed using deionized water several times until the solution became neutral in pH. The washed powders were then dried in the oven at 100 °C overnight before use for the following investigations. 

### 2.2. Material Characterization

The morphology of the obtained samples was observed by scanning electron microscopy (SEM, SU8000, Hitachi High-Technologies Co., Tokyo, Japan). The structural properties were analyzed using an X-ray diffractometer (XRD, DX-2600, Haoyuan Instrument Co., Ltd., Dan-dong, China). The coordination states of materials were inspected by solid-state nuclear-magnetic resonance (NMR, Avance 400, Bruker BioSpin GMBH, Rheinstetten, Germany) under 104.26 MHz, 79.5 MHz, and 105.84 MHz for ^27^Al, ^29^Si, and ^23^Na measurements, in which AlClO_4_, Si(CH_3_)_4_, and NaCl were used for the standard samples, respectively. The elemental composition was measured by X-ray fluorescence spectroscopy (XRF, NEX CG, Rigaku Co., Tokyo, Japan). The specific surface area was measured using BET (Flow Prep 60, Micromeritics Instrument, Norcross, GA, USA). The chemical bonding states were characterized by X-ray photoelectron spectrometry (XPS, K-Alpha, Thermo Fisher Scientific Inc., Waltham, MA, USA). 

### 2.3. Adsorption Experiments

The isotherm batch adsorption experiments were carried out by placing 0.1 g of powder sample in 10 mL of Sr^2+^, Co^2+^, and Cs^+^ standard solutions in various concentrations from 10 mg/L to 3000 mg/L, prepared using SrCl_2_, CoCl_2_, and CsCl. Isotherm adsorption was performed in a constant-temperature oscillator at 25 °C for 24 h to ensure complete sorption equilibrium. Note that the pH of the isotherm adsorption condition ranged from 5.3–4.4 with increasing the concentration of the standard solutions from 10 to 3000 mg/L. The filtrates were collected using the centrifugal technique, and the concentrations of Sr^2+^, Co^2+^, and the released Na^+^ ions were measured using an atomic absorption spectrometer (AAS, Perkin Elmer Analyst 100). Kinetic adsorption experiments were conducted by agitating 10 mL of Sr^2+^ and Co^2+^ solutions in 100, 1000, and 3000 mg/L with 0.1 g of powder sample at 25 °C. At appropriate time intervals, the filtrates were obtained via the centrifugal method and then analyzed by AAS. The powder samples after the adsorption experiments were collected for the XRF analyses. Finally, the competitive adsorption experiments were carried out using Sr^2+^, Co^2+^, and Cs^+^ ternary solutions in equal concentrations from 10 mg/L to 3000 mg/L to study the behavior of the three different ions. The experimental conditions were identical to that employed for the batch isotherm adsorption described above. 

### 2.4. Preliminary Immobilization Test

The samples after the batch adsorption experiment at 3000 mg/L Sr^2+^ and Co^2+^ were collected and dried at 100 °C overnight to obtain dried powders. Mineralization of the post-adsorption samples was carried out via heat treatment at 1300 °C for 2 h. The mineralized samples were characterized by XRD. A leaching test was applied following the International Stand ISO 6961-MCC 3 process [13]. The mineralized powders were placed in water with a surface area of solid sample to water ratio at 1, and stirred at 70 °C for 30 days. The concentration of the leaching ions was measured using AAS as a function of the reaction time.

## 3. Results and Discussion

### 3.1. Material Characterizations

Alkali activation was performed by soaking the metakaolin powders in NaOH solution for 1, 3, and 7 days at ambient temperature, and the samples were labeled BK1, BK3, and BK7, respectively. The elemental compositions of these alkali-activated samples in comparison with the original metakaolin are listed in Table 1. The metakaolin was mainly composed of SiO_2_ and Al_2_O_3_, where the Al/Si atomic ratio was 1.04 with slight amounts of TiO_2_, and other oxide (i.e., Fe_2_O_3_, ZrO_2_, and K_2_O). By increasing the soaking time, the impurity compounds were gradually dissolved by NaOH, while the amount of Na_2_O increased to 9.5%. The Al/Si ratios remained unvaried despite the soaking time. In addition, the specific surface area (S_BET_) increased from 20.3 m^2^/g to 56.4 m^2^/g for BK1, corresponding to the dissociation of the metakaolin structure. The surface area decreased to 48.9 m^2^/g as Na_2_O increased to 22.4% for BK3, showing that the aluminate and silicate units were connected through the condensation reaction, while Na^+^ ions played a key role as a binder [14]. The cation-exchange capability (CEC) taking into account the content of Na_2_O were 3.91, 6.89, and 6.99 meq/g for BK1, BK3, and BK7, respectively. The XRD patterns of metakaolin revealed a semi-crystalline structure with a broad band located at 15°–27°, while a slight amount of quartz and anatase TiO_2_ coexisted as impurities. BK1 and BK3 samples revealed similar broad features, however the location shifted to 19°–22° and 25°–28°, respectively (Figure 1). This indicated that the geopolymeric structure was modified due to the rearrangement of aluminosilicate framework, as the Na^+^ ions were intercalated during the condensation reaction. Obvious sharp XRD peaks associated to Linde Type A zeolite (LTA) and faujasite (FAU) were found in BK7. The extraordinarily high specific surface area (454.3 m^2^/g) was fairly consistent with the appearance of zeolite phases. SEM images of metakaolin, BK1, BK3, and BK7 are demonstrated in Figure 2. The metakaolin exhibited a sheet-like structure (Figure 2a) that was gradually destroyed in BK1 and BK3 (Figure 2b,c). The sheet-like feature completely disappeared for BK7 and the product morphology corresponded well to typical LTA and FAU zeolites [15].

### 3.2. Polymerization Process

To further understand polymerization in the alkali activation process, ^27^Al, ^29^Si, and ^23^Na NMR analyses were employed, and the results are shown in Figure 3. In metakaolin, Al exhibited AlO_4_ (Al^IV^), AlO_5_ (Al^V^), and AlO_6_ (Al^VI^) coordination, which was in agreement with the reported typical metakaolin structure [16]. Alkali activation effectively transformed Al into Al^IV^ coordination with resonance centered at 62 ppm due to the Na^+^ balanced Si-O-Al^IV^ [17]. The coordination of Al remained unchanged with increasing the soaking time. The ^29^Si NMR spectra of metakaolin exhibited broad resonance centered at −102 ppm, which was assigned to Q^4^(1Al). A resonance centered at around −80 ppm emerged along alkali activation, which was associated with Q^n^(mAl) (0 ≤ m ≤ n ≤ 4) [18]. The polymerization process was not yet completed in BK1, as a part of Si persisted in Q^4^(1Al) form as the pristine metakaolin. The Si in coordination with Q^4^(1Al) nearly disappeared and turned into Q^n^(mAl), showing that a higher amount of Al was substituted in the aluminosilicate-based polymeric framework. The ^29^Si NMR spectrum of BK7 became obviously sharp, and two splitting resonances located at −80 and −90 ppm contributed to the crystallized structure of FAU and LTA, respectively [19,20]. The ^23^Na resonances located on −4.2 and −3.3 ppm were found in BK1 and BK3, respectively, which were attributed to Na^+^ associated with Al-centered tetrahedral as a charge-balancing role that was a typical form of Na^+^ in alkali-activated aluminosilicate [21]. The Na resonance shifted to a higher frequency, indicating a decrease in coordination number and shortening in the average Na-O interatomic distance [19]. The ^27^Al and ^29^Si NMR investigations revealed that a higher amount of Al in participation within the polymeric framework may be the reason for the decrease in Na^+^ coordination number, which shortened the Na-O interatomic distance. Na^+^ resonance in BK7 obviously shifted to 2 ppm, which was generally observed in the hydrated Na^+^ in the zeolite structure [20,22]. 

### 3.3. Adsorption Capacity of Sr^2+^ and Co^2+^

The adsorption isotherms of Sr^2+^ and Co^2+^ on metakaolin and alkali-activated metakaolin according to the batch adsorption experiments are shown in Figure 4a,b. The adsorption capability of metakaolin was very low for Sr^2+^ nor Co^2+^ and was obviously enhanced by alkali activation treatment. The adsorption efficiency of Sr^2+^ basically increased with the soaking time, and that of Co was optimal for BK3. The adsorption isotherm characteristics are often evaluated through the Langmuir isotherm model (Equation (1)), as described in Equation (1).
(1)$$\frac{C_{e}}{q_{e}}=\frac{C_{e}}{q_{\max}}+\frac{1}{K_{L}q_{\max}}$$
where q_e_ is the equilibrium adsorption capacity at various concentrations, q_max_ is the maximal adsorption capacity, K_L_ is the constant adsorption equilibrium, and C_e_ is the equilibrium concentration of the target ion in the solution. According to the plots C_e_/q_e_ versus C_e_, as shown in Figure 4c,d, the Sr^2+^ and Co^2+^ adsorption isotherm data fitted the Langmuir model with correlation coefficients (R^2^) close to 1. The adsorption parameters according to the fitting results are listed in Table 2. 

The maximum adsorption capacity (q_max_) increased with the soaking time for both Sr^2+^ and Co^2+^. The values of q_max_ of Sr^2+^ were 2.13, 3.81, and 4.10 meq/g for BK1, BK3, and BK7, which were equivalent to 93.5, 167.5, and 180.2 mg/g, respectively. The Sr^2+^ adsorption capacity was basically proportional to the CEC value (Table 1). Note that the active surface area is commonly proposed to be an important parameter for adsorption application; nonetheless, S_BET_ of BK7 was about 15-fold higher than that of BK3, but q_max_ of BK7 was only slightly higher, indicating that the specific surface area was not the dominant factor determining the adsorption capacity of Sr^2+^. The adsorption capacities of Co^2+^ were 2.14, 4.02, and 3.30 meq/g for BK1, BK3, and BK7, equivalent to 63.1, 118.5, and 98.0 mg/g, respectively. The q_max_ of Co^2+^ and Sr^2+^ were generally identical for BK1, but quite different for BK3 and BK7. The q_max_ value of Sr^2+^ in BK7 was higher than that in BK3, but this was opposite for Co^2+^. The limit of adsorption capacity of Co^2+^ found in BK7 was suggested to be the molecular sieve effect due to the presence of LTA and FAU. The inherent pore apertures in LTA and FAU were 0.41 and 0.74 nm, respectively [23]. The hydrated radii of Sr^2+^ and Co^2+^ were 0.412 and 0.432 nm, respectively [24]. The small pore aperture size of the LTA phase may have imposed a transport barrier for the diffusion of Co^2+^ and obstruct the contact with the active sorption sites that eventually resulted in a decrease in adsorption capacity due to this geometric limitation. According to the adsorption isotherm results, BK7 showed a good adsorption efficiency to Sr^2+^; however, the LTA phase appeared to restrict the adsorption capacity to Co^2+^ due to the molecular sieve effect. BK3 exhibited a semi-crystalline structure that eliminated the molecular sieve effect; therefore, it showed a good adsorption ability and was considered an optimal adsorbent for the removal of Sr^2+^ and Co^2+^.

### 3.4. Adsorption Kinetics 

In this section, the adsorption kinetics of BK3 to Sr^2+^ and Co^2+^ were investigated to understand the sorption mechanism. Three different initial concentrations, namely, 100, 1000, and 3000 mg/L, were used, and the results are shown in Figure 5a,b. For the initial solution of 100 mg/L, Sr^2+^ and Co^2+^ adsorption was completed in a very short time. Two adsorption steps were clearly observed in the solutions of a high concentration. Very rapid adsorption was found in the first few minutes for both Sr^2+^ and Co^2+^, and then slowed down until reaching the adsorption equilibrium. The pseudo-first-order (PFO) and the pseudo-second-order (PSO) kinetic models were used to fit the adsorption kinetic data, as listed in Equations (2) and (3).
(2)$$\ln\left(\frac{q_{t}}{q_{e}-q_{t}}\right)=\mathrm{kt}$$
(3)$$\frac{t}{q_{t}}\;=\;\frac{t}{q_{e}}+\frac{1}{{\mathrm{kq}}_{e}{}^{2}}$$
where q_e_ and q_t_ are the adsorption capacities at equilibrium and at time t, respectively, and k is the rate constant corresponding to the PFO and PSO models. The linear graphs of ln(q_e_ − q_t_) and t/q_t_ against time (t) corresponding to the PFO and PSO models are shown in Figure 5c–f. The fitting parameters are listed in Table 3. The experimental data fitted the PSO model better based on the correlation coefficient (R^2^) of linear regression instead of the PFO model in comparison with their correction constants (R^2^). The PFO model considers that adsorption and desorption processes generally adapt to a high initial concentration, whereas the PSO model involves a complex function that is adapted for a low solution concentration [25]. However, the adsorption kinetics observed in the present work and the experimental data all fitted the PSO model, regardless of the initial concentration, implying that no obvious desorption occurred. The PSO model is based on the assumption that the rate-limiting-step is chemisorption, involving a sharing or exchange of electron between the adsorbent and adsorbate [25,26,27,28]. In addition, the rate constant was dependent on the initial concentration, which was another characteristic of the chemical exchange reaction [22]. Regarding the fitting parameters listed in Table 3, the rate constants (k) of Sr^2+^ adsorption derived from the PSO model were found faster than those of Co^2+^ adsorption. This may be related to the larger hydrated radii of Co^2+^ that led to a slower diffusion rate and retarded the reaction rate. Note that q_e_ obtained at 3000 mg/L (167.8 mg/g) was identical to the maximum adsorption capacity (q_max_) derived from the Langmuir isotherm (167.5 mg/g). Thus, the adsorption of Sr^2+^ primarily occurred at the surface of BK3, because the Langmuir model is based on the monolayer sorption on distinct localized sorption sites at the surface of the adsorbent. The equilibrium capacity (q_e_) for Co^2+^ at 3000 mg/L (122.4 mg/g) was similar to that calculated from the Langmuir isotherm (118.5 mg/g), showing that the sorption of Co^2+^ was also related to the chemisorption that occurred at the localized site on the BK surface. 

### 3.5. Sorption Mechanism

The adsorption behaviors of Sr^2+^ or Co^2+^ onto BK3 satisfied Langmuir adsorption and the pseudo-second-order (PSO) model for chemisorption. To further understand the chemisorption mechanisms, the released amount of Na^+^ ions at the batch adsorption experiments were measured to confirm if ion exchange occurred. The concentrations of Na^+^ (C_e_-Na) in comparison with the equilibrium adsorption capacities (q_e_) at various initial concentrations are shown in Figure 6. For Sr^2+^ adsorption, the concentrations of Na^+^ were generally proportional to the equilibrium adsorption capacities of Sr^2+^ (q_e_-Sr). This result showed that the sorption of Sr^2+^ was accompanied with a release of Na^+^, confirming that ion exchange was the main mechanism. The ion exchange between Na^+^ and Co^2+^ was not evident at concentrations below 1000 mg/L and became more significant at high concentrations. Thus, a certain amount of the Co^2+^ was removed from the solution via mechanisms independent of the ion exchange reaction, particularly at low initial concentrations. The relevant discussions are further described by XPS analyses in the next section. 

The chemical states of the post-adsorption samples, which were agitated at 3000 mg/L for 24 h, were collected and their XPS spectra, are shown in Figure 4. The Sr-3d core level XPS spectrum (Figure 7a) could be deconvoluted into two Gaussian separated peaks located at 134.1 and 135.9 eV, which were assigned to Sr-3d_5/2_ and Sr-3d_3/2_, respectively. The binding energies of these two characteristic peaks were observed in Sr containing oxide compounds [27,28], but were also found in Sr^2+^ ions replacing the Na^+^ ions in CHA-type zeolite [29]. Therefore, Sr-O bonds were formed. Sr^2+^ adsorption was confirmed to proceed via the ion-exchange mechanism. The Co-2p XPS spectrum shown in Figure 7b could be deconvoluted into four Gaussian peaks. The peaks at 781.2 and 797.1 eV were the main peaks of Co-2p_3/2_ and Co-2p_1/2_ of Co(OH)_2_ or Co(OH)^+^ accompanying the satellite peaks at 785.6 and 802.8 eV due to the simultaneous excitation of electrons and charge transfer from the ligand to metal [30,31,32,33]. The precipitation of Co(OH)_2_ is a pH-sensitive reaction that occurs at a pH above 7, whereas Co(OH)^+^ possibly exists at acid environments below 6 [34,35]. As the adsorption experiment was performed at an environment ranging from 5.7 to 4.4, the precipitation of Co(OH)_2_ should not be the reaction for the removal of Co^2+^ from the solution. Co^2+^ adsorbed onto the BK3 surface via surface complexation in the form of Co(OH)^+^ was suggested to be a sorption mechanism. In combination with the results shown in Figure 6b, the surface complexation preferentially occurred at the lower concentrations and was independent of the release of Na^+^ ions. As the hydroxyl groups on the BK3 surface were completely consumed, ion exchange, accompanying a dissociation of Na^+^ ions into the solution, was at a high concentration.

### 3.6. Competitive Adsorption of Sr^2+^, Co^2+^ and Cs^+^

The adsorption isotherms of BK3 using single ion and ternary ions solutions containing Sr^2+^, Co^2+^, and Cs^+^ and their adsorption isotherm are shown in Figure 8a,b. According to the plots C_e_/q_e_ versus C_e_, as shown in Figure 8c,d, the Sr^2+^, Co^2+^ and Cs^+^ adsorption isotherm data both in single and tenary solute solutions fitted the Langmuir model with correlation coefficients (R^2^) close to 1. The adsorption parameters are shown in Table 4. In a single-solute system, the maximum adsorption capacities (q_max_) of Sr^2+^, Co^2+^, and Cs^+^ were 167.5, 118.5, and 190.2 mg/g, equivalent to 3.82, 4.02, and 1.43 meq/g, respectively. The adsorption isotherm revealed that the alkali-activated metakaolin preferred the adsorption of divalent ions (Sr^2+^ and Co^2+^) prior to the monovalent Cs^+^. The maximum adsorption capacities (q_max_) of Sr^2+^, Co^2+^, and Cs^+^ in the ternary-solute system all decreased down to 76.6, 68.4, and 61.2 mg/g (equivalent to 1.74, 2.32, and 0.46 meq/g), indicating that adsorption was hindered by the presence of other ions. In particular, the decrease in adsorption capacity of Cs^+^ was more significant than that of Sr^2+^ and Co^2+^. The adsorption selectivity was in the order Co^2+^ > Sr^2+^ >> Cs^+^. The competitive adsorption behavior between Sr^2+^ and Cs^+^ in Na-based zeolites was strongly related to the Al/Si ratios [33]. In general, Cs^+^ has a higher sorption ability prior to Sr^2+^ in high-silica zeolites, [36,37] and Sr^2+^ has a higher sorption selectivity at alumina-rich zeolites [38,39]. The adsorption selectivity in monovalent and divalent is designated to dielectric theory, in which the decrease in Al/Si ratio lowers the charge density and dielectric constant of the framework that favors the exchange of monovalent cations [40,41]. In addition, the Al−Al distance distribution shifted to a lower value with increasing the Al/Si ratio, which allowed for bridging two ion-exchange sites for the exchange of divalent cations [36]. Those mentioned above concerned the ion exchange behaviors of Cs^+^ and Sr^2+^ in crystallized zeolites, but could also be adapted for the semi-crystalline BK3 sample. The adsorption capacities of Sr^2+^ and Co^2+^ both decreased in the ternary solute system, but the sum of their maximum adsorption capacities was almost the same as their corresponding q_max_ in a single-solute solution. Therefore, Sr^2+^ and Co^2+^ occupied the same adsorption sites, but were less affected by the presence of Cs^+^ ions.

### 3.7. Mineralization

To prevent secondary pollution, the adsorbed ions must be immobilized for waste management. The waste BK3 containing the adsorbed Sr^2+^ or Co^2+^ was calcined at 1300 °C for 2 h, and the XRD patterns of the as-calcined wasteforms are shown in Figure 9. The as-calcined Sr^2+^-containing wasteform was crystallized into the slawsonite (SrAl_2_Si_2_O_8_) phase, while the Co^2+^-containing wasteform remained in the amorphous state. Note that the volume of the waste was largely reduced, as demonstrated in Figure 9. A preliminary leaching test was performed by merging the calcined wasteforms in deionized water at 70 °C for 90 days, and no ions were detected. This result was consistent with the standard requirement of the leaching rate below 10^−5^–10^−6^/cm^2^ day. This preliminary leaching test indicated that Sr^2+^ or Co^2+^ were immobilized in the calcined wasteforms. 

## 4. Conclusions

Alkali-activated metakaolin was prepared at ambient temperature for 1, 3, and 7 days. Polymerization proceeded with the soaking time while the Na^+^ ions gradually intercalated into the as-obtained products and finally formed FAU and LTA after soaking for 7 days. The adsorption behavior of Sr^2+^ and Co^2+^ both fitted the Langmuir adsorption and the pseudo-second-order kinetics models corresponding to a rate-controlling step by chemisorption mechanism. Sr^2+^ adsorption was generally proportional to the content of Na^+^ in the product. However, the adsorption efficiency for Co^2+^ was limited by the presence of zeolite phases in BK7 due to the molecular sieve effect. The BK3 demonstrated an optimal adsorption ability for both Sr^2+^ and Co^2+^, and their maximum adsorption capacity reached 3.81 and 4.02 meq/g, respectively. The investigation further confirmed that Sr^2+^ adsorption in BK3 was carried out by the ion exchange mechanism, whereas Co^2+^ adsorption was achieved by ion exchange and partial surface complexation in the form of Co(OH)^+^ onto the surface of BK3. The alkali-activated metakaolin with a high Al/Si ratio (approximately 1) showed a good competitive adsorption on Sr^2+^ and Co^2+^ and less effective adsorption on Cs^+^. The adsorbed Sr^2+^ and Co^2+^ could be immobilized in the used adsorbent via heat treatment at 1300 °C for 2 h in the form of crystallized slawsonite (SrAl_2_Si_2_O_8_) and Co-consisting glass. The present work proposes an eco-friendly process using kaolinite as a raw material to prepare a highly efficient adsorbent for removing Sr^2+^ and Co^2+^, and the final waste product could be directly mineralized and vitrified via heat treatment to immobilize the Sr^2+^ and Co^2+^ ions with a high degree of volume reduction. The alkali-activated metakaolin exhibited a high adsorption efficiency to remove Sr^2+^ and Co^2+^ from the aqueous solution, so it is promising for decontamination application in nuclear wastewater.

## Figures and Tables

**Figure 1 polymers-14-00992-f001:**
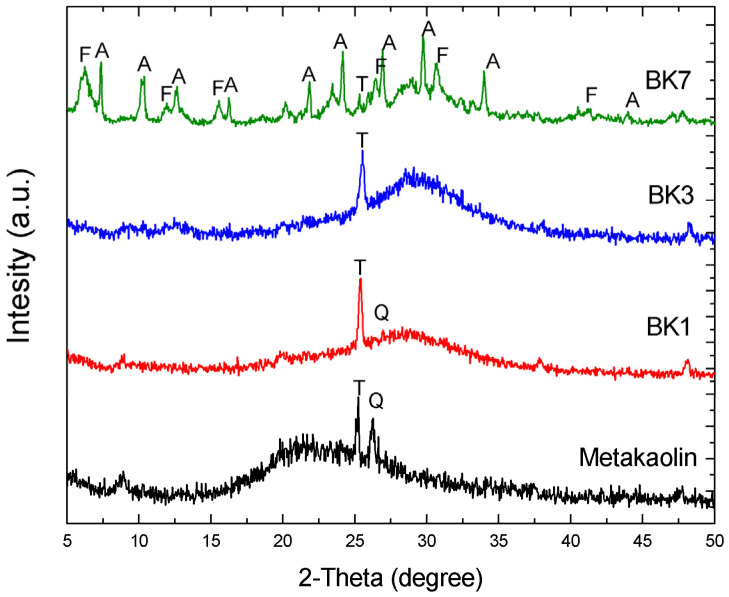
XRD pattern of metakaolin, BK1, BK3, and BK7. T: anatase TiO_2_; A: Linde type A zeolite; F: Faujasite.

**Figure 2 polymers-14-00992-f002:**
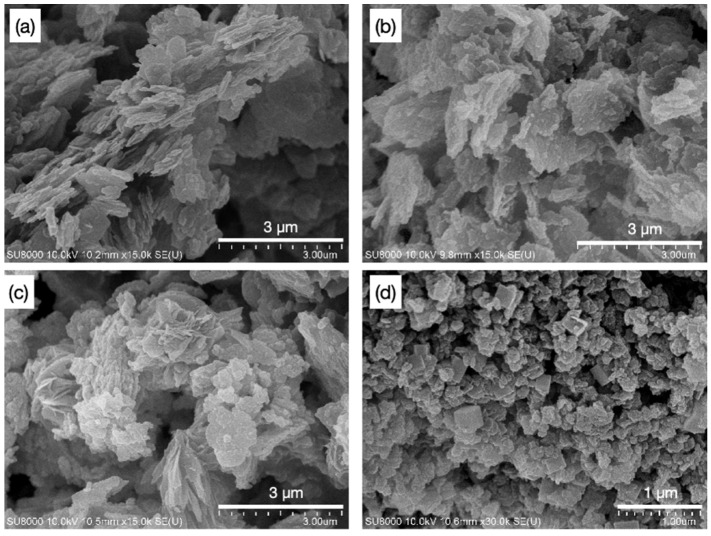
SEM images of (**a**) metakaolin, (**b**) BK1, (**c**) BK3, and (**d**) BK7.

**Figure 3 polymers-14-00992-f003:**
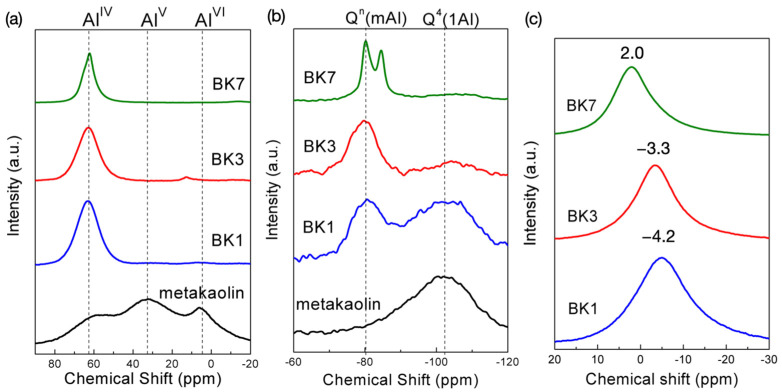
(**a**) ^27^Al, (**b**) ^29^Si, and (**c**) ^23^Na NMR spectra of BK1, BK3, and BK7.

**Figure 4 polymers-14-00992-f004:**
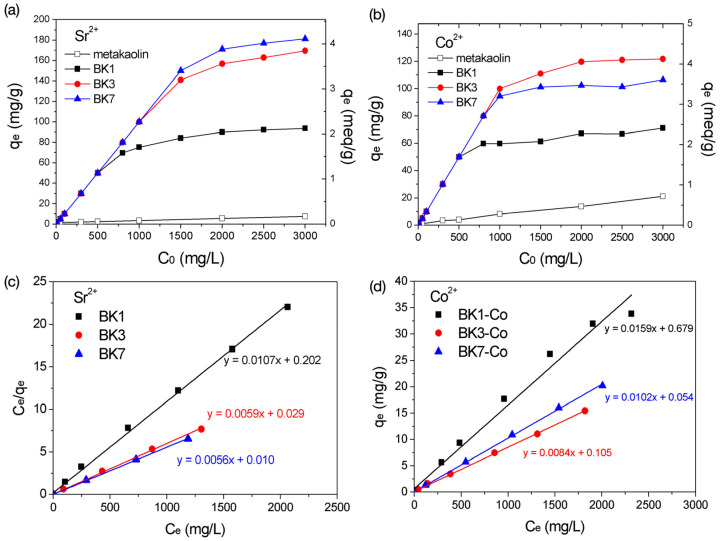
(**a**,**b**) Adsorption isotherms and (**c**,**d**) Langmuir plot of metakaolin and alkali-activated samples for Sr^2+^ and Co^2+^.

**Figure 5 polymers-14-00992-f005:**
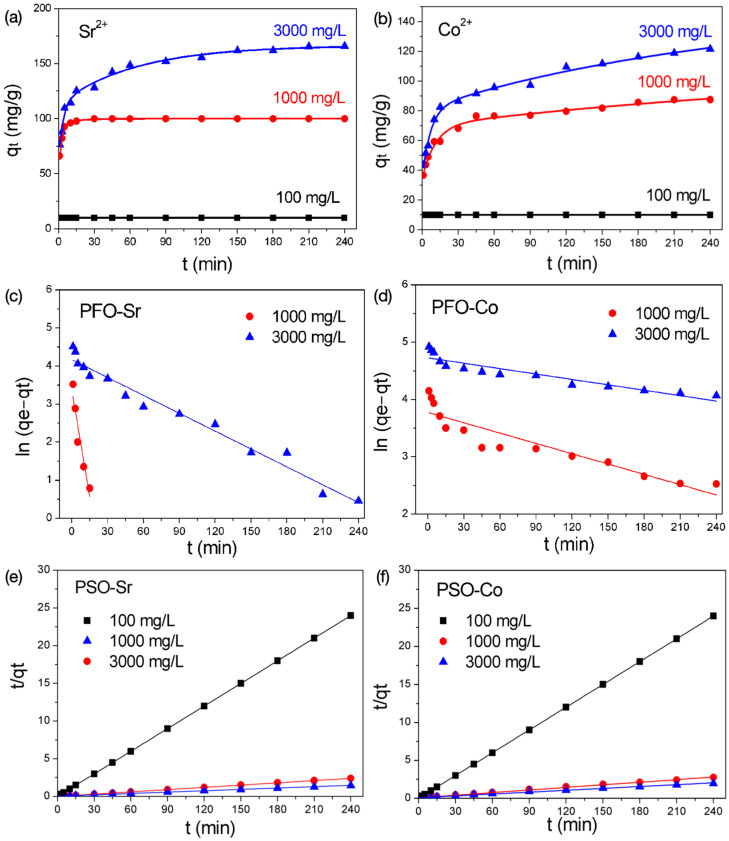
(**a**,**b**) Kinetic adsorption; (**c**,**d**) PFO model and (**e**,**f**) PSO plots of Sr^2+^ or Co^2+^.

**Figure 6 polymers-14-00992-f006:**
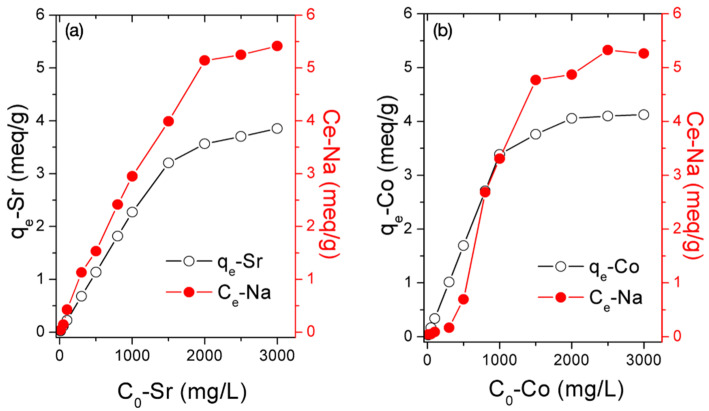
Adsorption−desorption of (**a**) Sr-Na and (**b**) Co-Na at various concentrations.

**Figure 7 polymers-14-00992-f007:**
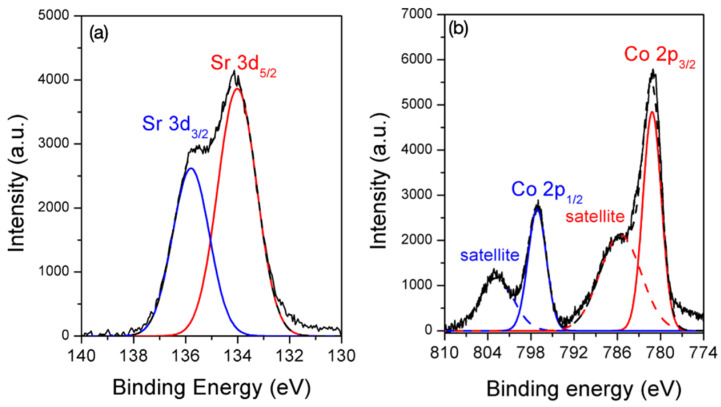
(**a**) Sr-3d and (**b**) Co-2p XPS spectra of post-adsorption BK3.

**Figure 8 polymers-14-00992-f008:**
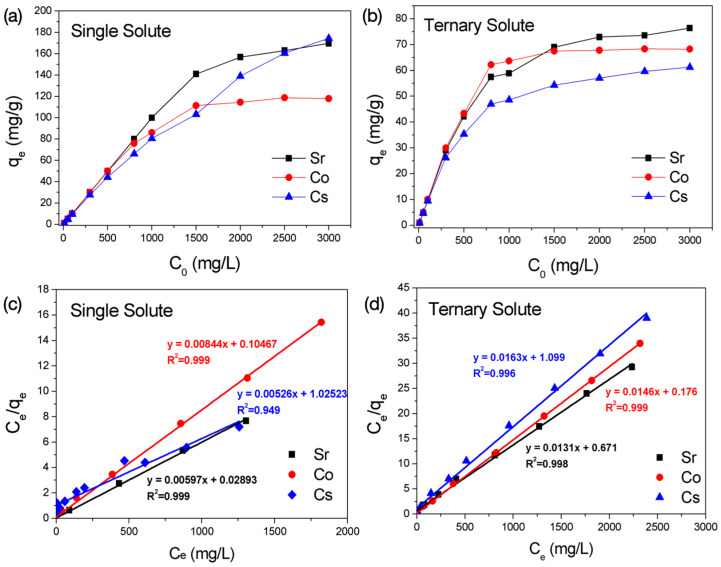
Adsorption isotherm of Sr^2+^, Co^2+^, and Cs^+^ in (**a**,**c**) single solute and (**b**,**d**) ternary solute solution.

**Figure 9 polymers-14-00992-f009:**
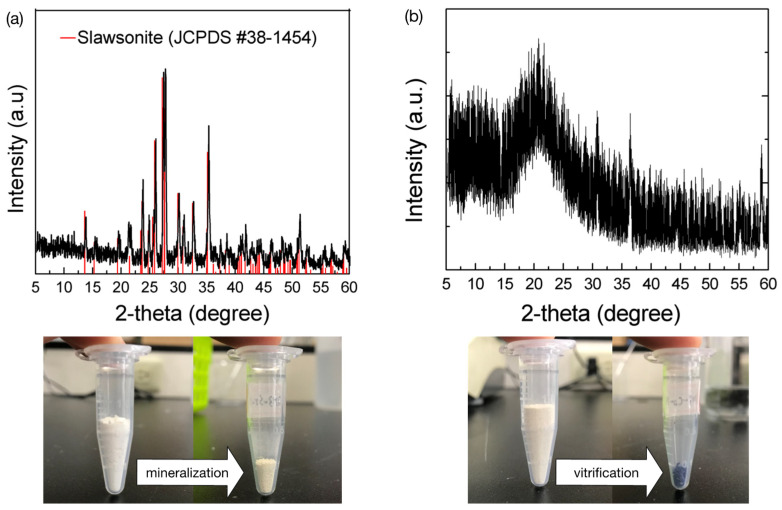
XRD patterns of BK3 containing the adsorbed (**a**) Sr^2+^ and (**b**) Co^2+^ calcined at 1300 °C for 2 h.

**Table 1 polymers-14-00992-t001:** Elemental composition and specific surface area (S_BET_) of metakaolin and the alkali-activated samples.

	Content (At.%)					CEC (meq/g)	Al/Si	S_BET_ (m^2^/g)
	SiO_2_	Al_2_O_3_	Na_2_O	TiO_2_	Other Oxides			
Metakaolin	56.5	29.4	-	9.9	4.1	-	1.04	20.3
BK1	58.9	27.9	9.5	3.1	0.5	3.91	0.95	56.4
BK3	50.6	24.1	22.4	2.6	0.3	6.80	0.95	48.9
BK7	49.8	23.4	24.7	1.7	0.4	6.99	0.94	454.3

**Table 2 polymers-14-00992-t002:** Adsorption parameters of BK samples from the Langmuir isotherm model.

Sample	R^2^	K_L_	q_max_ (mg/g)	q_max_ (meq/g)
Sr^2+^				
BK1	0.998	0.053	93.5	2.13
BK3	0.998	0.206	167.5	3.81
BK7	0.999	0.555	180.2	4.10
Co^2+^				
BK1	0.984	0.023	63.1	2.14
BK3	0.999	0.080	118.5	4.02
BK7	0.999	0.189	98.0	3.30

**Table 3 polymers-14-00992-t003:** Kinetic adsorption parameters of BK3 by PFO and PSO models.

Concentration	Sr^2+^			Co^2+^		
	k	q_e_	R^2^	k	q_e_	R^2^
**Pseudo-First-Order Model (PFO)**						
1000 mg/L	195.9	35.0	0.978	6	43.5	0.845
3000 mg/L	156.6	64.6	0.970	3.14	113.0	0.861
**Pseudo-Second-Order Model (PSO)**						
1000 mg/L	30.46	100.0	0.999	1.71	88.3	0.997
3000 mg/L	1.09	167.8	0.998	0.83	122.4	0.994

Unit of k: 10^−3^; q_e_: mg/g.

**Table 4 polymers-14-00992-t004:** Adsorption parameters of Sr^2+^, Co^2+^, and Cs^+^ in single solute and ternary solute solution from the Langmuir isotherm model.

Sample	R^2^	K_L_	q_max_ (mg/g)	q_max_ (meq/g)
**Single Solute**				
Sr^2+^	0.999	0.206	167.5	3.82
Co^2+^	0.999	0.080	118.5	4.02
Cs^+^	0.949	0.555	190.2	1.43
**Ternary Solute**				
Sr^2+^	0.998	0.019	76.6	1.74
Co^2+^	0.999	0.083	68.4	2.32
Cs^+^	0.996	0.014	61.2	0.46

## Data Availability

The data presented in this study are available on request from the corresponding author.

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
