# Peer review of "Highly Efficient Adsorption of Sr2+ and Co2+ Ions by Ambient Prepared Alkali Activated Metakaolin"

_polymers, 2022, doi:10.3390/polym14050992_

Round 1
Reviewer 1 Report
The manuscript should be supplemented with up-to-date references!
Lines 24-27: Please add references.
Lines 34-35: Please add references regarding zeolites, their well-known facts.
Line 45: Please rephrase, maybe it is precise to write “such as metakaolin, fly ash and slag and it can be used for heavy metals removal materials”
Line 105: Please rephrase “Pristine” with more appropriate like pure metakaolin. The main goal of calcination of kaolinite to produce metakaolin is to remove organic matter in the first line and some organic impurities.
Line 115: It is not proper to say microstructure of amorphous structure? It is more precise to write “XRD results reveals semi-crystalline geopolymer structure formation” since geopolymer materials can be described as materials with structures that represent a partial order in the domains of the structure. Since the structure is not completely decomposed and there are still peaks of residual mineral phases, mostly quartz, this order could be called semi-crystalline.
Line 115: For the first time I saw peaks of anatase with so high intensities in metakaolin after geopolymerization. I must confirm that is very unusual, can you please explain the high content of TiO2 in metakaolin it is almost 10%? Is it related to the geological site of taking samples?
The peaks of quartz are missing and it represents a leading secondary mineral in metakaolin.
Line 182: Yes, that comment can be correct, but in this case, in sample BK7 it is most like to have hemi and physic sorption doubled since this sample has the largest surface area.
The sorption studies are written very well and precisely.
Author Response
Dear reviewer,
We are grateful to the reviewer for their useful and practical comments for our manuscript. We have revised the manuscript according to the comments and highlighted the changes within the revised manuscript. The point-by-point responses to reviewers’ comments were listed below:
Lines 24-27: Please add references.
The relevant reference was added.
Lines 34-35: Please add references regarding zeolites, their well-known facts.
The relevant reference was added.
Line 45: Please rephrase, maybe it is precise to write “such as metakaolin, fly ash and slag and it can be used for heavy metals removal materials”
The context was modified according to the useful suggestions.
Line 105: Please rephrase “Pristine” with more appropriate like pure metakaolin. The main goal of calcination of kaolinite to produce metakaolin is to remove organic matter in the first line and some organic impurities.
The context has been corrected.
Line 115: It is not proper to say microstructure of amorphous structure? It is more precise to write “XRD results reveals semi-crystalline geopolymer structure formation” since geopolymer materials can be described as materials with structures that represent a partial order in the domains of the structure. Since the structure is not completely decomposed and there are still peaks of residual mineral phases, mostly quartz, this order could be called semi-crystalline.
We do agree with the reviewer’s comment that the geopolymer is a short-range-ordered structure that is different from the “amorphous” in definition. The relevant contexts mentioning structural properties by “amorphous” within the manuscript have been revised to “semi-crystalline”.
Line 115: For the first time I saw peaks of anatase with so high intensities in metakaolin after geopolymerization. I must confirm that is very unusual, can you please explain the high content of TiO2 in metakaolin it is almost 10%? Is it related to the geological site of taking samples?
It is true that such high content of TiO2 in metakaolin is unusual. The raw natural kaolinite was purchased from the American Clay Society, source by “Warren Country, Georgia, USA”. From their providing SDS data, TiO2 anatase should not be that much. Nevertheless, we have checked and repeated the experiments for many times and TiO2 is surely existed. Since the content of TiO2 was decreased with increasing the soaking time, and titanium is in form of oxide compound that should not affect the ion exchange behavior, we would leave the related discussions unmodified.
The peaks of quartz are missing and it represents a leading secondary mineral in metakaolin.
We have added the XRD patterns of metakaolin in the revised manuscript, quartz was indeed revealed in metakaolin and was almost vanished along the alkali activation treatment.
Line 182: Yes, that comment can be correct, but in this case, in sample BK7 it is most like to have hemi and physic sorption doubled since this sample has the largest surface area.
We agree that the high surface area may promote the physical adsorption, however, the specific surface area of BK7 is about 10 times higher than that of BK7 and the sorption capacity to either Sr or Co were not obviously enhanced. If the physical adsorption do occurred in BK7, it implies that the chemisorption efficiency should be quite low. As the qmax is basically proportional to the Na2O content, we would keep our original description the sorption mechanism in the alkali-activated metakaolin as the chemisorption, mainly by ion exchange and ignoring the possible physical sorption.
The sorption studies are written very well and precisely.
Thanks for your kind comment.

Reviewer 2 Report
This manuscript by Huang & Wu aims to assess the potential of alkali activated metakaolin for the adsorption of Co2+ and Sr2+. This manuscript is a classical adsorption manuscript in its shape, using interesting experimental techniques in order to validate the sorption mechanisms and durability of adsorption. The experiments have been diligently conducted and the manuscript is thoroughly written. I recommend minor modifications on this manuscript before publication. Specific comments are given below:
- Section 2.2 It lacks some useful details in this section, please complete.
- Section 2.3. What was the pH during adsorption experiments? and how was it controlled?
- Why did you not evaluate the CEC of adsorbents (Table 1)? It would be reinforce your further assumptions.
- P7L230-232 This assumption on possible desorption of Co2+ appears speculative as I'm not convinced that kinetic model may "anticipate" desorption , please clarify.
- Table 3 Please be homogeneous in the used units (qe in meq/g or mg/g for PSO and PFO)
- P9L246-250 This must be clarified explain, as you explained further (L268-271) that ion exchange occurs at high concentration for Co2+ please be more specific
- Section 3.6 Why didn't you used any model to fit the data of the ternary experiments?
- Such results were already obtained by other authors with other modified clay materials for example Ma et al. 2011 Desalination 276 or Thiebault et al. 2020 Materials 13, with same types of assumptions made on the competitive behavior, please extend the cited literature
Author Response
Dear reviewer,
We are grateful to the reviewer for their useful and practical comments for our manuscript. We have revised the manuscript according to the comments and highlighted the changes within the revised manuscript. The point-by-point responses to reviewers’ comments were listed below:
- Section 2.2 It lacks some useful details in this section, please complete.
We have added more details in this section.
- Section 2.3. What was the pH during adsorption experiments? and how was it controlled?
We did not control the pH condition for the adsorption experiments. The pH depend on the concentration of SrCl2, CoCl2 and CsCl standard solution that were ranging from 5.3 to 4.4 that the addition of the powder samples did not change the pH value. The description of pH was added in the relevant context in the revised manuscript.
- Why did you not evaluate the CEC of adsorbents (Table 1)? It would be reinforce your further assumptions.
Thanks for the useful suggestion. The CEC values of adsorbents were calculated and added in Table 1 in the revised manuscript. The contexts related to CEC were also added.
- P7L230-232 This assumption on possible desorption of Co2+ appears speculative as I'm not convinced that kinetic model may "anticipate" desorption, please clarify.
Thanks for the suggestion. We actually have no evidence that the desorption of Co2+ was occurred. Regarding to the value of qe (by kinetic adsorption experiment) and qmax (by Langmuir isotherm), the difference (122.4 vs. 118.5 mg/g) was not that large. We have thus removed the context concerning to the desorption.
- Table 3 Please be homogeneous in the used units (qe in meq/g or mg/g for PSO and PFO)
Thanks for the careful revision. The related data was all modified by the same unit mg/g.
- P9L246-250 This must be clarified explain, as you explained further (L268-271) that ion exchange occurs at high concentration for Co2+ please be more specific.
The discussions related to the sorption mechanism of Co2+ at low and high concentration were described in details in the section of XPS analysis.
- Section 3.6 Why didn't you used any model to fit the data of the ternary experiments?
The adsorption data fitting by Langmuir model was added in the latest manuscript.
- Such results were already obtained by other authors with other modified clay materials for example Ma et al. 2011 Desalination 276 or Thiebault et al. 2020 Materials 13, with same types of assumptions made on the competitive behavior, please extend the cited literature
Thanks for providing the information. We have read the suggested reference papers. The work of Ma et al (2011) investigated the phosphate modified montmorillonite and the competitive adsorption behavior was Cs>Co > Sr. The work of Thiebault et al. (2020) is to investigate the APTES modified Laponite to the adsorption properties of Cs, Co and Sr where Co was unaffected in competitive adsorption experiment. Their results were quite different from ours. It is hard to refer their work to explain our observed data. Thus we did not modify this section.
